

# Source site of internal solitary waves in the northern South China of westward shoaling thermocline

Gang Wang[1,2,3*], Yuanling Zhang[1,2,3], Chang Zhao[1,2], Dejun Dai[1,2,3], Min Zhang[1,2,3],

Fangli Qiao[1,2]

[1]Key Laboratory of Data Analysis and Applications, the First Institute of Oceanography, State Oceanic Administration, Qingdao 266061, China

[2]Laboratory for Regional Oceanography and Numerical Modeling, Qingdao National Laboratory for Marine Science and Technology, Qingdao 266237, China

[3]Key Laboratory of Marine Science and Numerical Modeling (MASNUM), the First Institute of Oceanography, State Oceanic Administration, Qingdao 266061, China

*Corresponding to*:Gang Wang (wangg@fio.org.cn)

**Abstract** This study use a three dimensional general circulation model, MITgcm with non-hydrostatic option, to study the source site of internal solitary waves (ISWs) observed in the northern South China Sea. Simulation reveals that besides Luzon Strait, ISWs in the northern SCS are also generated around Dongsha Islands and near the continental shelf break. It is one of the reasons that there are more wave package to the west of 120oE in SAR images, and even more to the west of 118oE. The generation process and propagation feature of ISWs in these source sites are described.

## 1 Introduction

South China Sea (SCS), who connects the Pacific via Luzon Strait in its northeast (Figure 1(a)), is one of the largest marginal seas of the world oceans. Figure 1(b) sketches the topography of northern portion of the SCS, in which two meridional running ridges are parallel in Luzon Strait: Lanyu Ridge (the eastern ridge, which is also known as Luzon Island Arc) and Hengchun Ridge (the western one). The SCS is also characterized by large amplitude internal solitary waves (ISWs) ubiquitously in the sea basin. In field observations at the northern SCS, ISWs are recorded once or twice each day (Duda, 2004) during spring tide, propagating westward from Luzon Strait into the SCS (Liu and Hsu, 2004; Zhao et al., 2004; Fang and Du, 2005; Guo and Chen, 2012).The double-ridge configuration in Luzon Strait favors the generation of ISWs when tidal current from the Pacific flows through the strait into the SCS.



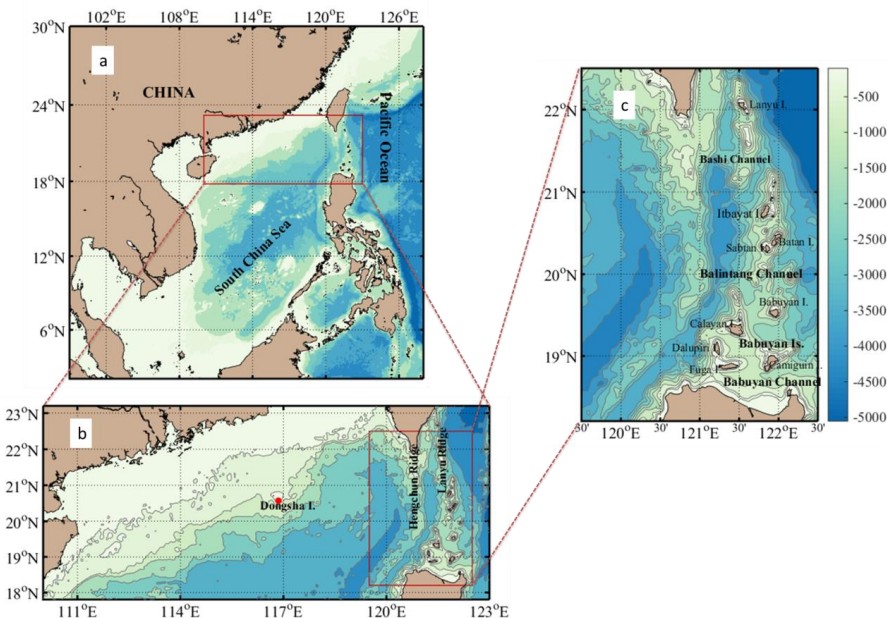

**Figure 1: The bathymetry of the South China Sea.**

Since the late 1970s, Synthetic Aperture Radars (SARs) have recorded thousands of ISWs all over
the SCS. Statistics based on SAR images (Liu and Hsu, 2004, 2009; Zhao et al., 2004; Fang and Du,
2005; Gan et al., 2007; Yang et al., 2008; Farmer et al., 2009; Cai et al., 2012) profiles the spatial
distribution of the ISWs in the SCS, in which wave crests concentrate in a band from Luzon Strait to
Hainan Island. The ISWs occurred in this band are believed to be originated directly from, or closely
related to the tidal current from the Pacific (Cai et al., 2002; Liu and Hsu, 2004; Zhao et al., 2004; Fang
and Du, 2005; Lien et al., 2005; Gan et al., 2007; Jan et al., 2007; Wang et al., 2010; Cai et al., 2012).
In the last 20 years, in situ observations have been carried out intensively in the northern SCS to
reveal the spatial and temporal features of ISWs beneath the sea surface. Cai et al. (2001) and Fang et
al. (2005) reported a mooring observation in 1998 which recorded several ISWs of amplitude over 100
m near the Dongsha Island (20.70$^o$N, 116.72$^o$E). The largest amplitude of those waves reaches up to
160 m. During the Asian Seas International Acoustics Experiment (ASIAEX) undertaken in April and
May 2001, twenty-one moorings deployed in the northeastern SCS recorded the shoaling of ISWs near
the continental shelf break (Orr and Mignerey, 2003; Liu and Hsu, 2004: Ramp et al., 2004; Yang et al.,
2004). Since 2005, observations related to the ISWs in the northern SCS increase noticeably (Yang et


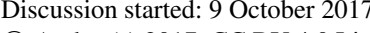


al., 2009; Xu et al., 2010a, 2010b; Klymak et al., 2006; Tian et al., 2006; Li and Farmer, 2011; Alford
et al., 2010; Liao et al., 2011; Si et al., 2010; Gao et al., 2010; Buijsman et al., 2012, 2014; Rainvelle et
al., 2013; Li et al., 2014; Huang et al., 2014; Pickering& Alford, 2015; Dong et al., 2015). These
observations reveal many details of spatial and temporal features on the ISWs in the northern SCS.
Table 1 is a portion of the observations since 1998.

**Table1: In situ observations on internal waves in the SCS since 1998**

| Reference | Observation Center | Time Span | Location |
|---|---|---|---|
| Cai, Gan et al., 2001<br>Fang, Shi et al., 2005<br>Orr et al., 2004; | Institute of Oceanology, CAS | 1998.05-1998.06 | 116°50.6'E; 20°21.3'N |
| Liu, Hsu, 2004;<br>Ramp et al., 2004;<br>Yang et al., 2004 | China, Kearo, Singapore et al., ASIAEX | Spring of 2000,<br>spring of 2001 | 117-119°E; 20-22.5°N |
| Xu et al., 2010a, 2010b | Institute of Oceanology, CAS | 2005.04- 2005.10 | 112°E; 19°35'N Wenchang station |
| Klymak et al., 2006 | R/V Revelle | 2005.04.15- 2005.05.15 | 119-120.5°E; 20-21°N |
| Yang et al., 2009 | Taiwan/U.S, VANS/WISE | 2005.04.29-2005.07.28<br>2005.11.02-2006.02.24 | 117°16.98'E; 21°36.87'N |
| Tian et al., 2006 | Ocean University of China R/V | 2005.10.04- 2005.10.16 | Meridional Transect at 120.5°E, Luzon Strait |
| Li, Farmer, 2011 | unknown | 2005, 2007 | 118.37-121.18°E; 20.39-21.22°N |
| Alford et al., 2010 | Taiwan and U.S., NLIWI | 2006.07-2007.05 | 116.5-121.8°E; 20.6-21.1°N |
| Rainville et al., 2013 | ONR etc. | 2007- | 23,000 profiles from gliders near Luzon Strait |
| Liao et al., 2011 | The Second Institute of Oceanography, SOA, China | 2008.04.25; 2008.09.26 | 120°30.33'E; 20°59.96'N |
| Gao et al., 2010 | Ocean University of China | 2008.08.24- 2008.09.03 | 121°E; 19.5-21°N |
| Sun et al. 2015 | SCOPE | 2008.07.01-2008.07.17 | 115.6°E, 22.0°N |
| | CHOICE-C | 2009.06-14-2008.07.07 | 116.66°E,22.19°N |
| Si et al., 2010 | Institute of Oceanology, CAS | 2009.06.24- 2009.06.25 | 117.5°E; 21°N, northeast to Dongsha Island |
| Li Zimu et al., 2014 | South China Sea Institute of Oceanology, CAS | 2010.09.04-2011.09.04 | 119°51'E; 20°33'N |
| Huang et al., 2014 | unknown | 2010.03-2010.08 | 119.1°E; 21.1°N |
| Pickering, Alford, 2015 | unknown | 2011 | 121°40.5'E; 20°31.4'N<br>121°1.8'E; 19°18'N |
| Dong et al., 2015 | Ocean University of China | 2012.10.02- 2012.10.04 | 118°24.62'E; 20°59.91'N |




In the recent 10 years, primitive equation ocean models become popular in investigating the
generation and propagation of both the ISWs and internal tides in the SCS (Table 2). Those models
include hydrostatic models such as ONFS (Chao et al., 2007), DieCAST (Du et al., 2008) and POM
(Jan et al, 2008); quasi-hydrostatic models such as HAMSOM (Song et al., 2010); and non-hydrostatic
models such as HAMSOM (Li Huang et al., 2011), ROMS (Buijsman et al., 2010b) and MITgcm (MIT
general circulation model, Marshall et al., 1997). Among these models the favorite one is probably the
non-hydrostatic MITgcm (Vlasenko et al., 2010; Wang et al., 2010; Li et al., 2011; Guo et al., 2011;
Guo et al., 2012; Vlasenko et al., 2012; Li Dan et al., 2012; Alfort et al., 2015; Wang et al., 2015; Wang
et al., 2016), which takes upon around half of the simulations that we listed in Table 2.

**Table 2: The simulations on internal waves in the SCS using primitive equation ocean models**

| Reference | Model | Nonhydrostatic | Initial T/S field | Model coverage |
|---|---|---|---|---|
| Chao et al., 2007 | ONFS | No | unknown | 116-123 °E, 17-23.5 °N |
| Du et al., 2008 | DieCAST | No | *H-uniform, 2D | 118-122 °E |
| Jan et al, 2008 | POM | No | H-uniform | 99.25-135.25 °E, 2.25-43.25 °N |
| Warn-Varnas et al, 2010 | unknown | Yes | NCOM, 2D | 117-123 °E |
| Buijsman et al, 2010 | ROMS | Yes | H-uniform, 2D | 20.58 °N , 117-126 °E |
| Song et al., 2010 | HAMSOM | Quasi | 2D | 20 °N , 115-130 °E |
| Li & Song et al., 2011 | HAMSOM | Yes | SODA 3D | 16-23 °N, 110-125 °E |
| Zhang et al., 2011 | SUNTANS | yes | H-uniform | 18-23 °N, 115-124 °E |
| Vlasenko et al., 2010 | MITgcm | Yes | H-uniform | 20-21 °N, 118-122.5 °E |
| Wang et al., 2010 | MITgcm | Yes | H-uniform, 2D | 20 °N , 118.5-123 °E |
| Li & Chen et al., 2011 | MITgcm | Yes | H-uniform, 2D | 16-16.3 °N, 110-118 °E |
| Guo et al., 2011 | MITgcm | Yes | H-uniform, 2D | 20-21.2 °N, 117-123 °E |
| Vlasenko et al., 2012 | MITgcm | Yes | H-uniform, 2D | 20.25-20.75 °N, 117-123 °E |
| Guo and Chen, 2012 | MITgcm | Yes | H-uniform | 20-21 °N, 118-122.5 °E |
| Li et al., 2012 | MITgcm Hallberg | Yes | H-uniform | 16-26 °N, 110-126 °E |
| Alfort et al., 2011 | Isopycnal model | unknown | H-uniform | 17-25 °N, 115-127.5 °E |
| Jan et al., 2012 | POM | No | H-uniform | 18-23 °N, 116.75-126.75 °E |
| Buijsman et al., 2012 | MITgcm | Yes | H-uniform, 2D | 20.6 °N , 120.25-122.25 °E |
| Wang, 2012 | POM | No | SODA 3D | 16-23 °N , 105.5-126 °E |
| BLK et al., 2013 | unknown | unknown | 2D | unknown |
| Buijsman et al., 2014 | MITgcm | Yes | H-uniform | Entire Luzon Strait |
| Pickering et al., 2015 | MITgcm, LZS | Yes No | H-uniform | Entire Luzon Strait Entire East Asian seas |
| Alfort et al., 2015 | MITgcm | Yes | H-uniform | 18-24 °N, 118-123 °E |





| Wang et al., 2015 | MITgcm | Yes | H-uniform | 18.5-22.5 °N, 114.5-124.5 °E |
| | | | | 18.5-22 °N, 116.2-123 °E |
| Wang et al., 2016 | MITgcm | Yes | H-uniform | 1.5-29.5 °N, 98.5-128.5 °E |

*H-uniform stands for horizontally uniform

Based on the observations (SAR images or field data), several mechanisms for the generation of the
ISWs in the northern SCS have been proposed. Tidal current from the Pacific (Hsu and Liu, 2000; Zhao
et al., 2004; Lien, 2005; Zhang et al., 2005; Fang et al., 2005; Zheng et al., 2008; Du et al., 2008), lee
waves to the west slope of the ridges (Warn-Varnas et al., 2010; Pinkel et al., 2012), Kuroshio (Hsu,
2000; Cai, 2003; Liu and Hsu, 2004; Yuan et al., 2006), and locally interacting nonlinear waves near
the thermocline (Zhao et al., 2004) are all possible candidates for triggering the ISWs. Besides
observation (SAR images and field experiments), numerical model is an important complementary tool
in studying the ISWs in the SCS. However, there are still many observed features of ISWs that have not
been explained by numerical simulation. For instance, simulation has not yet demonstrated how the A
waves (ISW packets rank-ordered with a large leading wave and some smaller followed) or B waves
(wave packets consisted of one single ISW) on northern shelf of the SCS are related to the tidal current
from the Pacific.
We noticed that all of the simulations using MITgcm listed in Table 2 start from a horizontally
homogenous density field, i.e., neglecting the spatial non-uniformity of stratification in the initial fields
but extending the temperature and salinity profiles at a point to the whole model domain. This approach
may cause bias for the numerical model to describe the generation and evolution of NIWs. For instance,
Zheng et al. (2007) discussed that the thermocline shoaling in the SCS forces the growth of ISWs. In
numerical experiments made by Buijsman et al. (2010), the westward shoaling of thermocline makes
the eastward solitons be 28% smaller than those westward. Besides, shoaling pattern of the thermocline
favors the growth of solitons. In this work, we set up an internal wave model with three dimensional
initial temperature and salinity climatology fields derived from WOA2013 (World Ocean Atlas 2013,
Locarnini et al., 2013; Zweng et al., 2013) as initial state, and study the source site of the ISWs in the
SCS based on numerical experiments.



## 2 Model configurations

A non-hydrostatic MITgcm code is set up to study the temporal and spatial features of ISWs in the
northern SCS. Compared with the simulations using MITgcm listed in Table 2, the current experiment
is distinctive in that the initial temperature and salinity fields are horizontally non-uniformly (Fig. 2).
This configuration makes the model more sensitive to the viscosity coefficient and temperature/salinity
diffusion coefficients since horizontal advection or diffusion also involves in the vertical processes.
The topography used in this work is the 1 arc-minute global relief model, ETOPO1 (Smith et al.,
1997; Amante and Eakins, 2011). And the internal wave model domain ranges from 110$^o$E to 123$^o$E in
longitude and 18$^o$N to 22.5$^o$N in latitude, covers the whole Luzon Strait, as in Fig.1(b) enclosed. Zonal
range from 110$^o$E to 123$^o$E is divided into 6000 grids with 210-m resolution. Meridional span of 4.5 arc
degree is divided into 800 grids with resolution of 620 m. The water column are divided into 60
z-coordinate layers, 5-to-10-m resolution in the upper 200 m, less than 30 m in the upper 440 m, and
then increases at a rate of about 1.2-fold until he maximum thickness of 700 m at the bottom layer. The
maximum water depth is set to 5086 m. At each of the three (east, north and south) open boundaries, 13
sponge layers covering 1.2 arc-degrees are set to relax the waves from reflecting. Their thickness
increased proportionally from innermost layer with normal grid to 40 km at the outermost boundary
layer. Table 3 lists the domain parameters, model resolutions, and the data source for the topography,
temperature/salinity field and external forcing.

**Table 3** Setups of the numerical model

| Topography | Initial T/S | Forcing current | Model domain | Resolution | | |
|---|---|---|---|---|---|---|
| | | | | x | y | z |
| ETOPO1 | WOA2013 Summer | TPXO7.1 11 components | 110-123$^o$E; 18 -22.5$^o$N | ~210 m, 6000 grids | ~620 m, 800 grids | 60 layers |


The initial temperature and salinity fields are derived from WOA2013. And we interpolate the three
dimensional climatological temperature and salinity in summer (JJA) into model grids.
The isotherm cross each meridional section shows a westward shoaling pattern. Meanwhile, a
subsurface tongue of high salinity centers at about 300 m depth intruding westward from Luzon Strait.





As a result, the isopycnic presents a westward shoaling trend. The vertical density gradient reaching
maximum 90 m depth around 120$^o$E and less than 50 m over the continental shelf.

Tidal current with 11 constituents (M2, S2, N2, K2, K1, O1, P1, Q1, MF, MM, M4) is derived from

TPXO7.1 (Egbert and Erofeeva, 2002) to force the model from the open boundaries. The running starts
from July 13, 2015, lasted one week. Figure 2 is the tidal current along the eastern boundary (123$^o$E) of
the model in the first four days of the simulation. Diurnal current is dominant while the semi-diurnal
component is less important. Meridional tidal current is stranger than zonal component, and in tidal
current in northern portion is stronger than southern portion.

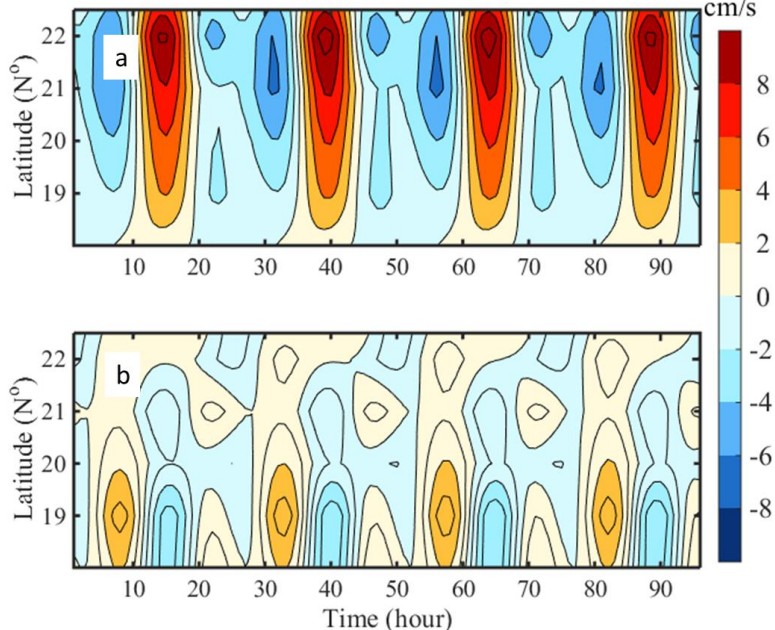


**Figure 2: Tidal current at the eastern boundary in the first four days of the simulation, (a) eastward   current**
**(positive to the Pacific) and (b) northward current.**

**3    Simulation and Discussion**
The roughness of sea surface derived from surface current ($u$,$v$) is a widely used approach to inverse
the internal wave's signature, i.e.,
$G = \sqrt{(\frac{\partial u}{\partial x})^2 + (\frac{\partial v}{\partial y})^2}$                                                                                    (1)





A similar approach is to replace *u* and *v* with sea surface displacement.

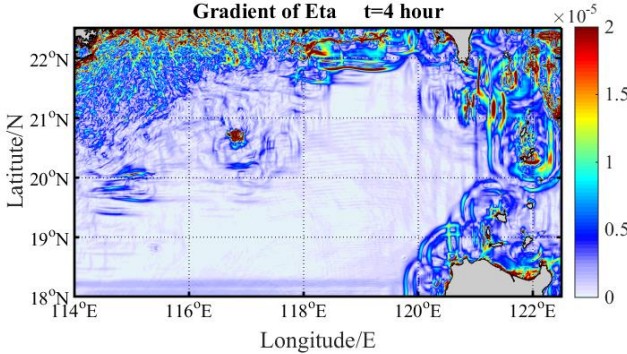

**Figure 3: Roughness of the sea surface elevation (color contour, unit: 1/s) at 4 hour. Signatures of internal**
**waves are found in Luzon Strait, around Dongsha Island and near the northern shelf break.**

Figure 3 gives gradient of sea surface displacement after 4 hours' simulation. In the northern SCS,
internal waves are found in three regions: in Luzon Strait, around the Dongsha Island and near the shelf
break. All of the three regions have steep slopes along the tidal current direction, which favors the
disturbance of current and therefore the isotherms. The roughness of sea surface over continental shelf
is also great, but the active waves there dissipated quickly and no long wave crest is formed.
We'll discuss the generation and propagation of internal waves in these three regions.
**3.1 Luzon Strait**
**3.1.1 In Luzon Strait**
Most of the studies on ISWs in the northern SCS indicates that they are generated in Luzon Strait or
around (Hsu and Liu, 2000; Zhao et al., 2004; Lien, 2005; Zhang et al., 2005; Fang et al., 2005;
Zheng et al., 2008; Du et al., 2008; Warn-Varnas et al., 2010; Pinkel et al., 2012; Hsu, 2000; Cai, 2003;
Liu and Hsu, 2004; Yuan et al., 2006). Simulations on internal tides further determined that the
primary generation sites are along the Lanyu Ridge and weaker generation along the Hengchun Ridge
(Niwa and Hibiya, 2004; Zhang and Fringer, 2006; Jan et al., 2008). It also showed that the Hengchun
Ridge have the potential either to reduce the energy of westward propagating internal waves (Chao et
al., 2007) or amplify westward propagating waves owing to resonance of the semidiurnal internal tide
trapped between the ridges (Buijsman et al., 2010a). Zheng et al. (2008) speculated from SAR images





that the west slope of Hengchun Ridge is a high possible source sites for long-crest transbasin ISWs
in the northern SCS.
In our simulation, disturbance of baroclinic current and the fluctuation of isotherms confirm that
Luzon Strait is an important source site of the ISWs over the northern shelf of the SCS. To the west of
the strait (around 121$^{o}$E) the density shows a high horizontal gradient. Current and temperature
pattern are disturbed when the tidal current flows over the Hengchun Ridge. Therefore, the ridge
around 121$^{o}$E is a main source site of ISWs over the northern shelf of the SCS.
In less than 3 hours after the starting of the simulation, strong current perturbation radiated out
from the ridges in Luzon Strait. The perturbation releases both eastward and westward propagating
waves. In northern portion of the strait, the westward waves are mainly generated over Hengchun
Ridge; in southern portion, the internal waves generate in the water channels between the islands,
which are parts of the Lanyu Ridge. Due to the sparsely distributed islands, the internal waves
generated in the channels are in forms of ripples rings (Fig. 4a). And they become a wave crest when
reaching the Hengchun Ridge. The wave crest (Fig. 4b) parallels to Hengchun Ridge, which is the
typical pattern in SAR image (Wang et al., 2010). Comparing the sea surface roughness at section
21.6$^{o}$N (Fig. 4c) and the oscillation of isotherms (Fig. 4d), we find that waves with small half-width in
the ocean are always corresponding to peaks of roughness at the sea surface. At those points, the wave
current also shows strong convergence (Fig. 4d).

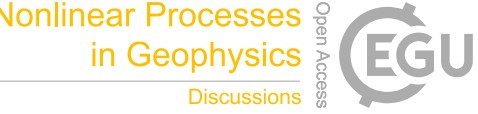

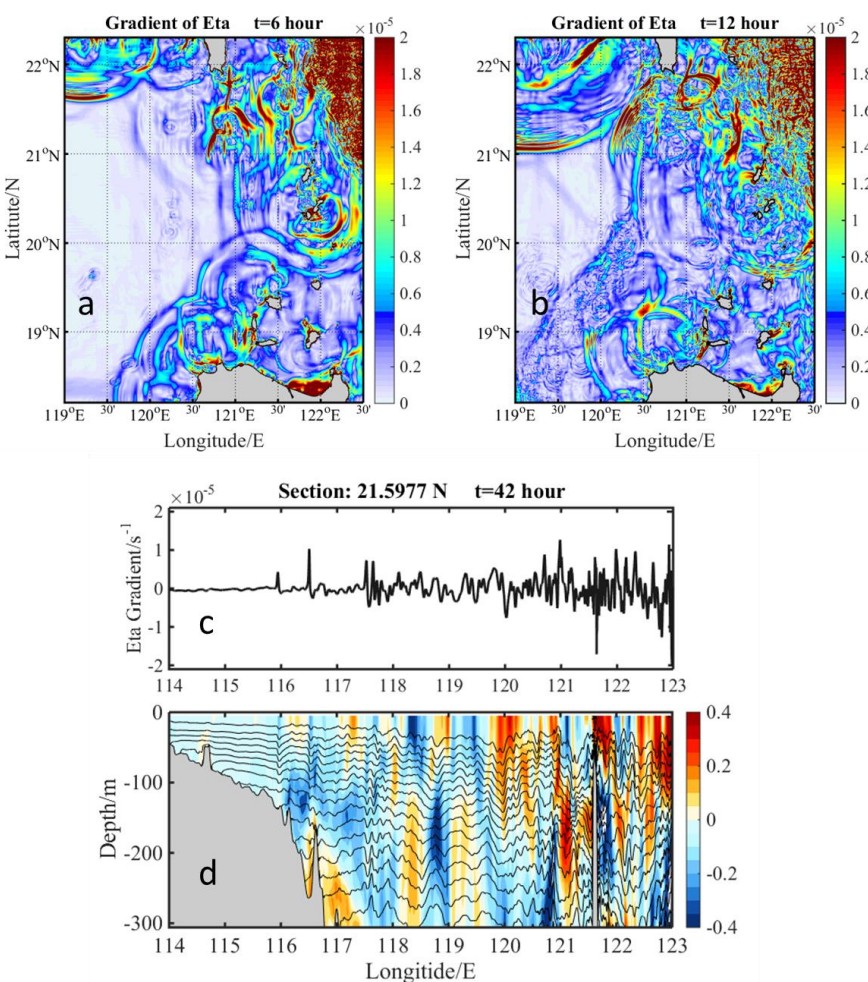

**Figure 4: Roughness of sea surface at (a) 6 h and (b) 12 h, (c) at 42 h across 21.6oN section, and (d) the**
**baroclinic current (colored, unit: m/s) and isotherms (black lines) across 21.6oN section.**

**3.1.2 Diffraction and Interference**
Small islands sparsely distributed in the Luzon Strait (see Fig.1c). Between them are shallow water
channels, where are the potential source sites of ISWs.
Ebbesmeyer et al. (1991) and Bole et al. (1994) deduced from satellite images that the ISWs in the
northern SCS are originated in the 4-km-wide narrow channel between Batan Island and Sabtang Island.
Inspecting over 100 SAR images, Hsu et al. (2000) found a crest of ISWs which extend 140 km long



but is only 50 km from those two islands. It indicates that the source site of this wave crest should be
extended out of the narrow channel. Tracking back from the propagation direction of the ISWs near
Dongsha Island, Cai et al. (2001) inferred that they are generated between Luzon Island and Fuga
Island (see Figure 1). Ramp et al. (2012) found ISWs between the ridges just to the south tip of the
Taiwan Island, indicating that the origin of ISWs in the northern SCS are not limited in the southern
portion of Luzon Strait. Huang et al. (2014) deduced detailed source sites for A waves and B waves: the
former is likely generated at the area south of the Batan Island, while the latter is possibly generated at
the area south of Itbayat Island and south of the Batan Island.

In our simulation the ripple rings radiate from narrow channels in Luzon Strait (Fig. 4). The widths

of these water channels are comparable with the wavelength of internal tides, and they serve as source
sites for ISWs. Due to diffraction, the ripple rings extended into the shadows of islands and interference
with each other, as the Huyghens' principle states. The longest wave crest in southern portion of Luzon
Strait is developed from the ripple rings out of the water channels. When propagating to the Hengchun
Ridge, those ripples form a long crest, paralleling to the ridge. Therefore, we cannot deduce the source
site of ISWs in Luzon Strait just by tracking back a straight line perpendicular to the wave crest.

For most of the simulation on the internal waves in Luzon Strait, the meridional grids are often

compromised to a resolution no less than 1 km to save the computing resource. In narrow channels they
are not fine enough to describe diffraction, and then may fail to characterize the long crest near the
strait.

### 3.2 Around Dongsha Island

Nonlinear internal waves around Dongsha Island are found in less than 4 hours after the starting of
simulation. Since Dongsha Island is more than 600 km away from the boundary, it is not likely that
these nonlinear internal waves are generated in Luzon Strait first and then propagated here in such a
short period. Therefore, barotropic tides with speed O(100 m/s) is the possible generator of those
waves.





**3.2.1 Reflection, diffraction, and interference**
In the gradient of sea surface elevation, ripple rings of reflected waves were found propagating
away from the Dongsha Island. The amplitude of these waves is about 30 m near the island (Fig. 5).

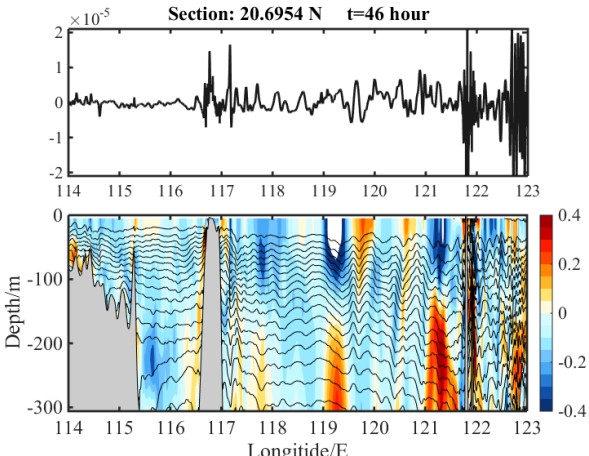


**Figure 5: Roughness of sea surface at 46 h across 20.7oN section (upper panel), the baroclinic current**
**(colored, unit: m/s) and isotherms (black lines) across that section.**

Due to the small size of Dongsha Island, diffraction of westward ISWs occurred around it. Waves
come into the western shadow part of the island after diffracted from north and south tips, and
interference take place behind the island. Figure 6 gives four snapshots of a long wave crest meeting
the Dongsha Island. At 36 h (Fig. 6a), two long wave crests propagate approaching the east boundary
of Dongsha Island. Wave crests are separated by the island into two branches (Fig. 6b). Reflection is
also happened but the eastward signal is very weak. With phase speed of about 1.3 m/s, these waves
diffract around the island (Fig. 3c). In a semi-diurnal cycle, another crest propagate approaching the
island (Fig. 3d).

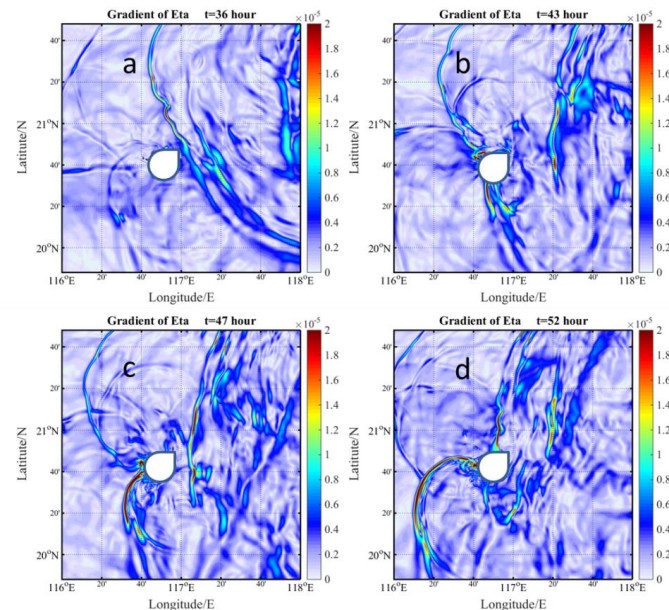

**Figure 6: Roughness of the sea surface elevation (color contour, unit: 1/s) near Dongsha Island at four snapshots, (a) 36 h, (b) 43 h, (c) 47 h, and (d) 52 h.**

**3.2.2 A wave and B wave**

Duda et al. (2004) classified the ISWs near Dongsha Island into two categories: A waves (type-a packet, which have one initial large wave and arrives the observation site at roughly the same time in each day), and B waves (type-b packets, which are less regular in amplitudes and wave timing, and arrive Dongsha Island at 25 to 26 hours interval). Zhao et al., (2004) also found internal wave packets in SAR images of two different types: a single-wave ISW packet containing only one ISW with/without an oscillating tail, and a multiple-wave ISW packet composed of a group of rank-ordered ISWs.

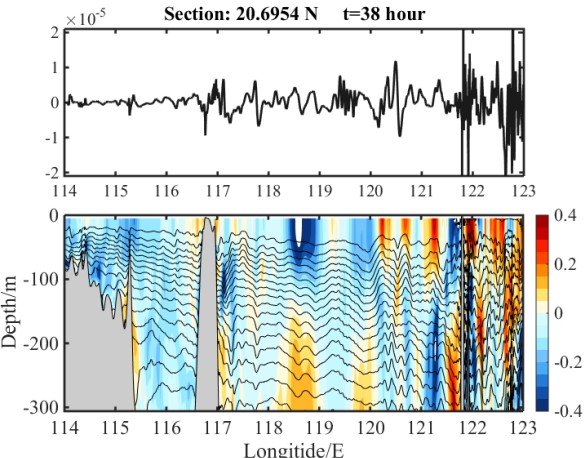

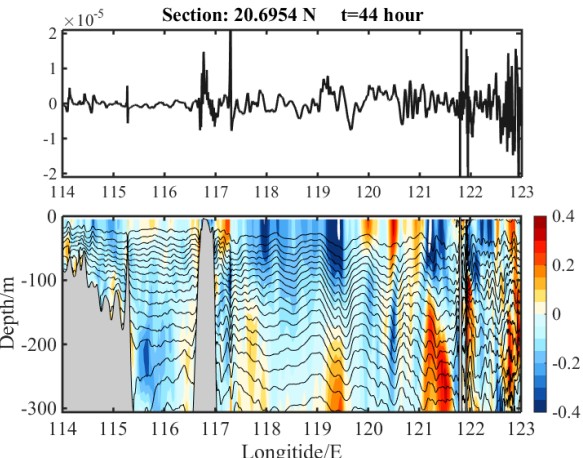

**Figure 7: The baroclinic current (colored, unit: m/s) and isotherms (black lines) across 20.7oN section,at 38 h**
**and 44 h, respectively.**

The propagation of A waves and B waves could reach a speed about 323 cm/s in the deep basin and
222 cm/s over the continental slope (Ramp et al. 2010). Calculating from a theoretical model, Alford et
al. (2010) discussed that A waves travel 5%−10% faster than B waves before they reach the continental
slope, while B waves travel 8%−12% faster than A waves on the upper continental slope.
Using the SUNTANS model of a nonhydrostatic simulation, Zhang et al. (2011) discussed that A
and B waves arise from the steepening of semidiurnal internal tides that are generated due to strong
barotropic flow over ridges in Luzon Strait. A wave is stronger in the southern portion of the Luzon



Strait because diurnal internal tidal beams enhanced the amplitude of the semidiurnal A waves. B wave
is stronger in the northern portion where the distance between the two ridges in the strait fits for
semidiurnal internal tidal resonance. The orientation of the ridges produces large A waves that
propagate into the southern portion of the western SCS basin and stronger B waves that propagate into
the northern portion. In Cai et al. (2012), A waves were generated around the time of the spring tides in
Luzon Strait and on the strong side of the diurnal inequality, while B waves were generated on the
weak side of the diurnal inequality. Note that Zhang et al. (2011) thought B wave is generated by
semidiurnal tides while Cai et al. (2012) believed it being generated by diurnal tides, the generation
mechanism for B waves is still in doubt. From the SAR images between 1995 and 2001, Zheng (2007)
found that only about 22% of ISWs in northern SCS occurred east to 118$^\circ$E. They are all single-wave
ISW packet containing only one ISW (Zhao 2004). Zhao (2004) suspected that some of the ISWs in the
northern SCS may not be generated in Luzon Strait, but locally generated in the thermocline by the
nonlinear interaction of internal tide beams from at about 120$^\circ$E.

The observation from October 2010 to September 2011 at center part (119$^\circ$51'E, 20$^\circ$33'N) of

Luzon Strait seems support that A waves and B waves are from different sources. In the observation
lasted almost one year, 105 solitary waves are recorded; all of them are B waves (Li et al., 2014).

Fig. 6 shows two wave crests reach the Dongsha Island at about semi-diurnal period. They are

originated from different sites. The first crest generated at the northern shelf of the SCS, while the
second crest generated in Luzon Strait. It is possible that A waves or B waves are not generated in
Luzon Strait but near the continental shelf break.
**3.3 Near Continental shelf break**
In statistics of ISWs imaged by SAR (e.g., Zhao et al., 2004; Zheng et al., 2007; Gan et al., 2007; Yang
et al., 2009), the wave crests to the west of 120$^\circ$E distributed much denser than to the east, and even
more denser to the west of 118$^\circ$E. For example, from the SAR images between 1995 and 2001, Zheng
(2007) found that only about 22% of ISWs in northern SCS occurred east to 118$^\circ$E.

Yang et al., (2004) believed that ISWs can be generated around shelf break by internal tide. Duda

et al. (2004) found that there are two categories of transbasin ISWs in the northern SCS: generated in
the Luzon Strait by tide or Kuroshio current, or generated at or near the continental shelf break by





transbasin waves and/or the diurnal tide.

When internal waves in Luzon Strait propagate westward into the SCS, increasing of nonlinear

effect enforces them into ISWs. During their way up to the shelf slope, due to the westward shoaling of
thermocline, their phase speed slows down. Therefore, there are more ISWs in shower regions. Another
reason for the non-homogenous distribution of wave crests is that some ISWs are generated just over
the slope by barotropic tides.

Besides the internal waves generated in Luzon Strait, internal waves near the shelf break also

become discernable no later than four hours after the starting of the simulation. Our simulation found
that the ISWs leave the Luzon Strait at about one diurnal cycle. Those waves found near the continental
shelf break seem not related to those internal waves generated in the Luzon Strait. In fact, only
barotropic wave (tidal current) is fast enough to cover four arc degrees in four hours. That is to see, the
internal waves near the shelf break is more likely being generated by barotropic tide rather than internal
tide.
**3.3.1 Source sites near the shelf break**
Simulation shows that internal waves generated over the slope form ripple rings and propagate leaving
the source site. The most prominent sea surface gradient is found about 2 degrees west to the south tip
of Taiwan Island (Fig. 7a). And another one is found south-east to the Hainan Island (Fig. 7b).


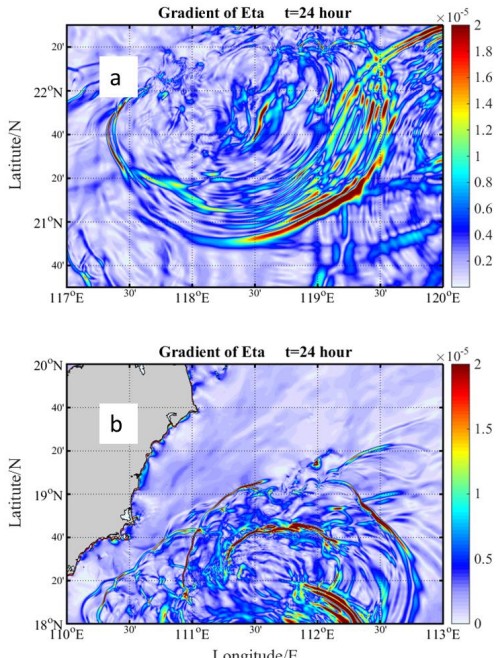


**Figure 7: Roughness of the sea surface (color contour, unit: 1/s) near northern continual shelf of SCS (a) west**

**to the south tip of Taiwan Island and (b) south-east to the Hainan Island.**


Non-uniformity of the model's initial state is not necessary for the generation of internal waves near
the shelf. In another simulation started from uniformly initial state, we also find internal waves
generated several hours from the starting of the simulation.
**3.3.2 Shape of the ISWs**
In the simulation, both depression and elevation ISWs are found near the shelf break. We also find a
snapshot when the isotherms in upper layers show elevation while those in deep layers are depression
(Fig. 8). It is a typical mode two ISW, which is not often found in the SCS. In the previous studies,
mode two ISWs had even been found at about 117$^o$E in a zonal section (Yang, 2009) and between the
ridges just to the south tip of the Taiwan Island (Ramp et al., 2012).
The largest amplitude of ISWs is commonly found near the pycnocline. In our simulation, however,
the ISWs near the shelf break reach their largest amplitude near the bottom (at about 200-m depth). The



corresponding baroclinic current implies that the ISWs in this region are mode two ISWs. Amplitude of
the mode two ISWs reach maximums on both sides of the pycnocline.

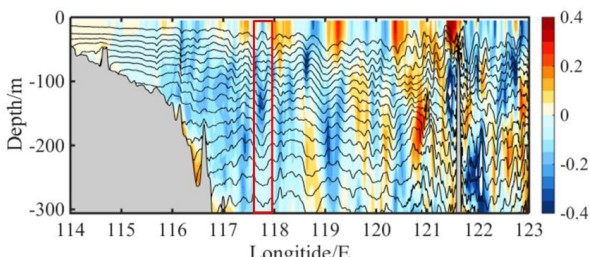


**Figure 8: The baroclinic current (colored, unit: m/s) and isotherms (black lines) across 21.6oN section, at 50**
**h.**

The internal waves fission over the shelf into several packets. Therefore, we find a big depression

wave followed by a pack of A waves over the slope, as Fig. 9 shows.

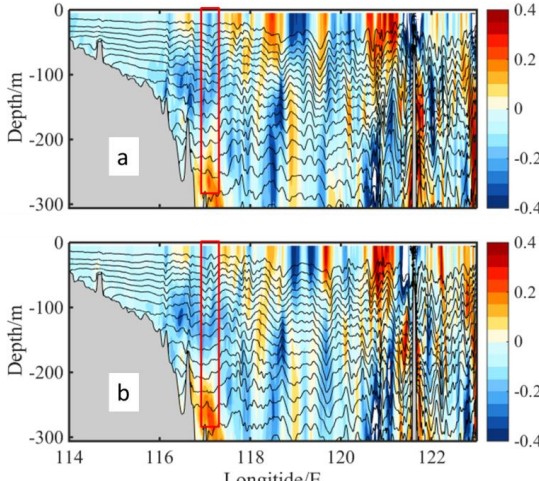


**Figure 9: The baroclinic current (colored, unit: m/s) and isotherms (black lines) across 21.6ᵒN section, at (a)**
**36 h, and (b) 38 h.**

**4. Conclusion**
We set up a non-hydrostatic MITgcm model to study the generation of ISWs in the northern SCS in
non-uniformly initial temperature/salinity fields. In the simulations, the propagation of the ISWs across

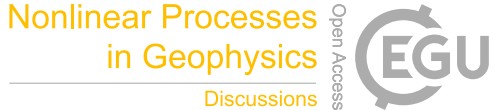

321 the SCS is similar to the schematic figure illustrated by Simmons et al. (2011). Our simulation,

322 however, reveals detailed source sites and evolution of ISWs. We find that the source sites of the ISWs

323 in the northern SCS are not limited in Luzon Strait. The shelf break and Dongsha Island are also

324 important source sites. And, the ISWs generated in these sites are likely being excited by barotropic

325 tide current from the Pacific rather than by internal tides.

326  In south portion of Luzon Strait, the internal waves generated and propagated out in the form of

327 ripple rings. Those rings form a long crest when they reach the Hengchun Ridge.

328  Simulation find that the wave crests near Dongsha Island comes from two areas: the northern shelf

329 break and Luzon Strait. A waves and B waves often observed near Dongsha Island may come from

330 these two areas, respectively.

331  Previous studies use the horizontal uniformly stratification as the initial state of numerical

332 simulation. It is reasonable when study the generation mechanism of the internal waves, but may have

333 some deficiency in practice. Firstly, it will take the model a longer time to reach the state as that of the

334 real ocean. Therefore, it is difficult to compare the simulation with real-time observations. Secondly,

335 the horizontal uniformly stratification comes from one profile of temperature and salinity, which is

336 often taken from deep-water region. Considering the westward shoaling of the stratification in the SCS,

337 the deviation caused by the stratification bias is expected. We believe that start the simulation from

338 non-uniformly stratification will reduce the possible bias caused by the westward shoaling thermocline

339 in the SCS and shorten the time for the model to get a steady state. Therefore, the simulation is likely to

340 yield internal waves more similar to the observations, and is thus applicable for the prediction of

341 internal waves in the SCS.

343 **Acknowledgments** G. Wang is supported by the National Key Research and Development Program of

344 China grant nos. 2016YFC1401407 and 2016YFB0201100, the National Natural Science Foundation

345 of China under grant no. 41476024; F. Qiao, D. Dai and Y. Zhang are supported by the

346 NSFC-Shandong Joint Fund for Marine Science Research Centers of China grant no. U1406404 and

347 the International Cooperation Project of Indo-Pacific Ocean Environment Variation and Air-Sea

348 Interaction of contract no. GASI-03-IPOVAI-05.






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
