# Peer review of "Source site of internal solitary waves in the northern South China of westward shoaling thermocline"

_Nonlinear Processes in Geophysics, 2017_

## Referee Comment (RC1) · E. Pelinovsky (Referee) · 22 Oct 2017

The paper is well written. It contains important results demonstrating the places where internal waves are generated. All conclusions are obtained with use of MIT code. I may recommend this paper for publication.

Minor comment. Internal waves in the South China Sea are a subject of intense studies. I can recommend to cite in Introduction following publications:

1. Grimshaw R., Talipova T., Pelinovsky E., Kurkina O. Internal solitary waves: propagation, deformation and disintegration. Nonlin. Processes Geophys. 2010, vol. 17,

[Figure]

No. 6, 633 – 649.

2. Talipova T.G., Pelinovsky E.N., Kharif Ch., Modulational instability of long internal waves of moderate amplitudes in stratified and horizontally inhomogeneous ocean. JETP Letters, 2011, vol. 94, No 3, 199-203.

3. Kurkina O., Talipova T., Soomere T., Giniyatullin A., Kurkin A., Kinematic parameters of internal waves of the second mode in the South China Sea. Nonlinear Processes in Geophysics, Nonlin. Processes Geophys. 2017, vol. 24, 645–660.

---

## Referee Comment (RC2) · Anonymous Referee #2 · 23 Oct 2017

This manuscript discusses simulations of the South China Sea performed with the MIT GCM in the non-hydrostatic configuration. The technical english (starting with the title and abstract) makes the paper difficult to assess, and would make it even more difficult for readers to get any detailed information from it. The graphics, on the other hand are quite good, with figures typically showing a proxy for sea surface roughness or slices in one direction and depth. The aim of the manuscript, to the extent I could make one out from the abstract, is to identify ISW generation sites. This is presented in brief subsections for some of the sites, along with other behaviour the authors find interesting (and a reader would as well). I found the discussion of the various points to be brief, difficult to follow due to poor technical English, and often bereft of quantitative information. In order to be a meaningful contribution, the manuscript requires a thorough proof reading and at least a partial application of the scientific method (I give two examples below). I am sympathetic to the difficulty in presenting the results of large numerical models, and there is room in the literature for a descriptive study, but even here more quantitative information (e.g. wave propagation speeds) is needed. I would like to see this manuscript published, but it needs some work on strengthening its scientific content, and a large amount of work on polishing up its presentation.

Language: I give ten examples of problems with the technical english and one request for longer captions. These are representative only and an exhaustive list would run much, much longer. However, most of these would be easy to catch by a proofreading for language only.

1) The title doesn't make sense as written

2) In the abstract "packets" should replace "package"

3) Line 57 how can hydrostatic models be used to model ISWs which are possible only due to a balance between nonlinearity and dispersion?

4) Line 86 "westward propagating", "eastward propagating" not eastward or westward

5) Line 93. There is only one MIT GCM, hence "The nonhydrostatic MIT GCM"

6) 100-105: I presume 13 sponge layers means a sponge layer that extends over 13 points? It would be good to have some supporting literature to suggest that this is enough to damp effectively without reflection. But at the very least please clarify what 13 refers to.

7) Line 130 what does "inverse the internal wave's signature" mean?

8) Line 167 "ripple rings" not "ripples rings"

9) Line 174 why is there an o after 21.6 and before N? In some captions this appears to correctly be the degree symbol, while in others it is the incorrect "o". This error occurs

in the text as well.

10) Line 178 is missing a verb

11) The figure captions are pretty short. They should tell the reader what they should pay attention to.

Material: 1) Given how complex the initial conditions are, I wonder what would happen if the model was initialized with the present initial conditions without tides. Or in other words how much of the wave generation is due to unbalanced initial conditions? Even a back of the envelope estimate would put the reader at ease.

2) Line 205: surely an estimate of the ISW propagation speed can be used to change "it is unlikely" to "it is certain"

3) Figure 8 needs at least one, perhaps two, lower panels to show the details of the waves in question. Is this an effect of shoaling, or purely due to the details of the stratification?

4) All the cross-sections involving depth show very long waves because they cover a large geographical area. Is there one in particular that has a clear ISW? This would be useful to identify.

5) While I appreciate that the more realistic thermocline is a good idea, science typically involves a null hypothesis, and I really wish the authors had run a test case with a horizontally uniform thermocline in order to demonstrate how their results are novel.